# Lower bounds on the robustness to adversarial perturbations

**Jonathan Peck**[1,2], **Joris Roels**[2,3], **Bart Goossens**[3], **and Yvan Saeys**[1,2]

[1]Department of Applied Mathematics, Computer Science and Statistics, Ghent University, Ghent, 9000, Belgium
[2]Data Mining and Modeling for Biomedicine, VIB Inflammation Research Center, Ghent, 9052, Belgium
[3]Department of Telecommunications and Information Processing, Ghent University, Ghent, 9000, Belgium

## Abstract

The input-output mappings learned by state-of-the-art neural networks are significantly discontinuous. It is possible to cause a neural network used for image recognition to misclassify its input by applying very specific, hardly perceptible perturbations to the input, called *adversarial perturbations*. Many hypotheses have been proposed to explain the existence of these peculiar samples as well as several methods to mitigate them, but a proven explanation remains elusive. In this work, we take steps towards a formal characterization of adversarial perturbations by deriving lower bounds on the magnitudes of perturbations necessary to change the classification of neural networks. The proposed bounds can be computed efficiently, requiring time at most linear in the number of parameters and hyperparameters of the model for any given sample. This makes them suitable for use in model selection, when one wishes to find out which of several proposed classifiers is most robust to adversarial perturbations. They may also be used as a basis for developing techniques to increase the robustness of classifiers, since they enjoy the theoretical guarantee that no adversarial perturbation could possibly be any smaller than the quantities provided by the bounds. We experimentally verify the bounds on the MNIST and CIFAR-10 data sets and find no violations. Additionally, the experimental results suggest that very small adversarial perturbations may occur with non-zero probability on natural samples.

## 1 Introduction

Despite their big successes in various AI tasks, neural networks are basically black boxes: there is no clear fundamental explanation how they are able to outperform the more classical approaches. This has led to the identification of several unexpected and counter-intuitive properties of neural networks. In particular, Szegedy et al. [2014] discovered that the input-output mappings learned by state-of-the-art neural networks are significantly discontinuous. It is possible to cause a neural network used for image recognition to misclassify its input by applying a very specific, hardly perceptible perturbation to the input. Szegedy et al. [2014] call these perturbations *adversarial perturbations*, and the inputs resulting from applying them to natural samples are called *adversarial examples*.

In this paper, we hope to shed more light on the nature and cause of adversarial examples by deriving lower bounds on the magnitudes of perturbations necessary to change the classification of neural network classifiers. Such lower bounds are indispensable for developing rigorous methods that increase the robustness of classifiers without sacrificing accuracy. Since the bounds enjoy the theoretical guarantee that no adversarial perturbation could ever be any smaller, a method which increases these lower bounds potentially makes the classifier more robust. They may also aid model selection: if the bounds can be computed efficiently, then one can use them to compare different

models with respect to their robustness to adversarial perturbations and select the model that scores the highest in this regard without the need for extensive empirical tests.

The rest of the paper is organized as follows. Section 2 discusses related work that has been done on the phenomenon of adversarial perturbations; Section 3 details the theoretical framework used to prove the lower bounds; Section 4 proves lower bounds on the robustness of different families of classifiers to adversarial perturbations; Section 5 empirically verifies that the bounds are not violated; Section 6 concludes the paper and provides avenues for future work.

## 2 Related work

Since the puzzling discovery of adversarial perturbations, several hypotheses have been proposed to explain why they exist, as well as a number of methods to make classifiers more robust to them.

### 2.1 Hypotheses

The leading hypothesis explaining the cause of adversarial perturbations is the *linearity hypothesis* by Goodfellow et al. [2015]. According this view, neural network classifiers tend to act very linearly on their input data despite the presence of non-linear transformations within their layers. Since the input data on which modern classifiers operate is often very high in dimensionality, such linear behavior can cause minute perturbations to the input to have a large impact on the output. In this vein, Lou et al. [2016] propose a variant of the linearity hypothesis which claims that neural network classifiers operate highly linearly on certain regions of their inputs, but non-linearly in other regions. Rozsa et al. [2016] conjecture that adversarial examples exist because of *evolutionary stalling*: during training, the gradients of samples that are classified correctly diminish, so the learning algorithm "stalls" and does not create significantly flat regions around the training samples. As such, most of the training samples will lie close to some decision boundary, and only a small perturbation is required to push them into a different class.

### 2.2 Proposed solutions

Gu and Rigazio [2014] propose the Deep Contractive Network, which includes a smoothness penalty in the training procedure inspired by the Contractive Autoencoder. This penalty encourages the Jacobian of the network to have small components, thus making the network robust to small changes in the input. Based on their linearity hypothesis, Goodfellow et al. [2015] propose the *fast gradient sign* method for efficiently generating adversarial examples. They then use this method as a regularizer during training in an attempt to make networks more robust. Lou et al. [2016] use their "local linearity hypothesis" as the basis for training neural network classifiers using *foveations*, i.e. a transformation which selects certain regions from the input and discards all other information. Rozsa et al. [2016] introduce *Batch-Adjusted Network Gradients* (BANG) based on their idea of evolutionary stalling. BANG normalizes the gradients on a per-minibatch basis so that even correctly classified samples retain significant gradients and the learning algorithm does not stall.

The solutions proposed above provide attractive intuitive explanations for the cause of adversarial examples, and empirical results seem to suggest that they are effective at eliminating them. However, none of the hypotheses on which these methods are based have been formally proven. Hence, even with the protections discussed above, it may still be possible to generate adversarial examples for classifiers using techniques which defy the proposed hypotheses. As such, there is a need to formally characterize the nature of adversarial examples. Fawzi et al. [2016] take a step in this direction by deriving precise bounds on the norms of adversarial perturbations of arbitrary classifiers in terms of the curvature of the decision boundary. Their analysis encourages to impose geometric constraints on this curvature in order to improve robustness. However, it is not obvious how such constraints relate to the parameters of the models and hence how one would best implement such constraints in practice. In this work, we derive lower bounds on the robustness of neural networks directly in terms of their model parameters. We consider only feedforward networks comprised of convolutional layers, pooling layers, fully-connected layers and softmax layers.

# 3 Theoretical framework

The theoretical framework used in this paper draws heavily from Fawzi et al. [2016] and Papernot et al. [2016]. In the following, $\|\cdot\|$ denotes the Euclidean norm and $\|\cdot\|_F$ denotes the Frobenius norm. We assume we want to train a classifier $f : \mathbb{R}^d \to \{1, \dots, C\}$ to correctly assign one of $C$ different classes to input vectors $\boldsymbol{x}$ from a $d$-dimensional Euclidean space. Let $\mu$ denote the probability measure on $\mathbb{R}^d$ and let $f^\star$ be an oracle that always returns the correct label for any input. The distribution $\mu$ is assumed to be of bounded support, i.e. $\mathrm{P}_{\boldsymbol{x} \sim \mu}(\boldsymbol{x} \in \mathcal{X}) = 1$ with $\mathcal{X} = \{\boldsymbol{x} \in \mathbb{R}^d \mid \|\boldsymbol{x}\| \leq M\}$ for some $M > 0$.

Formally, adversarial perturbations are defined relative to a classifier $f$ and an input $\boldsymbol{x}$. A perturbation $\boldsymbol{\eta}$ is called an *adversarial perturbation* of $\boldsymbol{x}$ for $f$ if $f(\boldsymbol{x} + \boldsymbol{\eta}) \neq f(\boldsymbol{x})$ while $f^\star(\boldsymbol{x} + \boldsymbol{\eta}) = f^\star(\boldsymbol{x})$. An adversarial perturbation $\boldsymbol{\eta}$ is called *minimal* if no other adversarial perturbation $\boldsymbol{\xi}$ for $\boldsymbol{x}$ and $f$ satisfies $\|\boldsymbol{\xi}\| < \|\boldsymbol{\eta}\|$. In this work, we will focus on minimal adversarial perturbations.

The *robustness* of a classifier $f$ is defined as the expected norm of the smallest perturbation necessary to change the classification of an arbitrary input $\boldsymbol{x}$ sampled from $\mu$:

$$\rho_{\mathrm{adv}}(f) = \mathrm{E}_{\boldsymbol{x} \sim \mu}[\Delta_{\mathrm{adv}}(\boldsymbol{x}; f)],$$

where

$$\Delta_{\mathrm{adv}}(\boldsymbol{x}; f) = \min_{\boldsymbol{\eta} \in \mathbb{R}^d} \{\|\boldsymbol{\eta}\| \mid f(\boldsymbol{x} + \boldsymbol{\eta}) \neq f(\boldsymbol{x})\}.$$

A *multi-index* is a tuple of non-negative integers, generally denoted by Greek letters such as $\alpha$ and $\beta$. For a multi-index $\alpha = (\alpha_1, \dots, \alpha_m)$ and a function $f$ we define

$$|\alpha| = \alpha_1 + \cdots + \alpha_n, \qquad \partial^\alpha f = \frac{\partial^{|\alpha|} f}{\partial x_1^{\alpha_1} \dots \partial x_n^{\alpha_n}}.$$

The *Jacobian matrix* of a function $\boldsymbol{f} : \mathbb{R}^n \to \mathbb{R}^m : \boldsymbol{x} \mapsto [f_1(\boldsymbol{x}), \dots, f_m(\boldsymbol{x})]^T$ is defined as

$$\frac{\partial}{\partial \boldsymbol{x}} \boldsymbol{f} = \begin{bmatrix} \frac{\partial f_1}{\partial x_1} & \cdots & \frac{\partial f_1}{\partial x_n} \\ \vdots & \ddots & \vdots \\ \frac{\partial f_m}{\partial x_1} & \cdots & \frac{\partial f_m}{\partial x_n} \end{bmatrix}.$$

## 3.1 Families of classifiers

The derivation of the lower bounds will be built up incrementally. We will start with the family of linear classifiers, which are among the simplest. Then, we extend the analysis to Multi-Layer Perceptrons, which are the oldest neural network architectures. Finally, we analyze Convolutional Neural Networks. In this section, we introduce each of these families of classifiers in turn.

A *linear classifier* is a classifier $f$ of the form

$$f(\boldsymbol{x}) = \arg \max_{i=1,\dots,C} \boldsymbol{w}_i \cdot \boldsymbol{x} + b_i.$$

The vectors $\boldsymbol{w}_i$ are called *weights* and the scalars $b_i$ are called *biases*.

A *Multi-Layer Perceptron* (MLP) is a classifier given by

$$f(\boldsymbol{x}) = \arg \max_{i=1,\dots,C} \mathrm{softmax}(\boldsymbol{h}_L(\boldsymbol{x}))_i,$$

$$\boldsymbol{h}_L(\boldsymbol{x}) = g_L(\boldsymbol{V}_L \boldsymbol{h}_{L-1}(\boldsymbol{x}) + \boldsymbol{b}_L),$$

$$\vdots$$

$$\boldsymbol{h}_1(\boldsymbol{x}) = g_1(\boldsymbol{V}_1 \boldsymbol{x} + \boldsymbol{b}_1).$$

An MLP is nothing more than a series of linear transformations $\boldsymbol{V}_l \boldsymbol{h}_{l-1}(\boldsymbol{x}) + \boldsymbol{b}_l$ followed by non-linear activation functions $g_l$ (e.g. a ReLU [Glorot et al., 2011]). Here, $\mathrm{softmax}$ is the softmax function:

$$\mathrm{softmax}(\boldsymbol{y})_i = \frac{\exp(\boldsymbol{w}_i \cdot \boldsymbol{y} + b_i)}{\sum_j \exp(\boldsymbol{w}_j \cdot \boldsymbol{y} + b_j)}.$$

This function is a popular choice as the final layer for an MLP used for classification, but it is by no means the only possibility. Note that having a softmax as the final layer essentially turns the network into a linear classifier of the output of its penultimate layer, $\boldsymbol{h}_L(\boldsymbol{x})$.

A *Convolutional Neural Network* (CNN) is a neural network that uses at least one convolution operation. For an input tensor $\mathbf{X} \in \mathbb{R}^{c \times d \times d}$ and a kernel tensor $\mathbf{W} \in \mathbb{R}^{k \times c \times q \times q}$, the discrete convolution of $\mathbf{X}$ and $\mathbf{W}$ is given by

$$(\mathbf{X} \star \mathbf{W})_{ijk} = \sum_{n=1}^{c} \sum_{m=1}^{q} \sum_{l=1}^{q} w_{i,n,m,l} x_{n,m+s(q-1),l+s(q-1)}.$$

Here, $s$ is the *stride* of the convolution. The output of such a layer is a 3D tensor of size $k \times t \times t$ where $t = \frac{d-q}{s} + 1$. After the convolution operation, usually a bias $\boldsymbol{b} \in \mathbb{R}^k$ is added to each of the feature maps. The different components $(\mathbf{W} \star \mathbf{X})_i$ constitute the feature maps of this convolutional layer. In a slight abuse of notation, we will write $\mathbf{W} \star \mathbf{X} + \boldsymbol{b}$ to signify the tensor $\mathbf{W} \star \mathbf{X}$ where each of the $k$ feature maps has its respective bias added in:

$$(\mathbf{W} \star \mathbf{X} + \boldsymbol{b})_{ijk} = (\mathbf{W} \star \mathbf{X})_{ijk} + b_i.$$

CNNs also often employ *pooling layers*, which perform a sort of dimensionality reduction. If we write the output of a pooling layer as $\mathbf{Z}(\mathbf{X})$, then we have

$$z_{ijk}(\mathbf{X}) = p(\{x_{i,n+s(j-1),m+s(k-1)} \mid 1 \leq n, m \leq q\}).$$

Here, $p$ is the pooling operation, $s$ is the stride and $q$ is a parameter. The output tensor $\mathbf{Z}(\mathbf{X})$ has dimensions $c \times t \times t$. For ease of notation, we assume each pooling operation has an associated function $I$ such that

$$z_{ijk}(\mathbf{X}) = p(\{x_{inm} \mid (n, m) \in I(j, k)\}).$$

In the literature, the set $I(j, k)$ is referred to as the *receptive field* of the pooling layer. Each receptive field corresponds to some $q \times q$ region in the input $\mathbf{X}$. Common pooling operations include taking the maximum of all inputs, averaging the inputs and taking an $L_p$ norm of the inputs.

## 4 Lower bounds on classifier robustness

Comparing the architectures of several practical CNNs such as LeNet [Lecun et al., 1998], AlexNet [Krizhevsky et al., 2012], VGGNet [Simonyan and Zisserman, 2015], GoogLeNet [Szegedy et al., 2015] and ResNet [He et al., 2016], it would seem the only useful approach is a "modular" one. If we succeed in lower-bounding the robustness of some layer given the robustness of the next layer, we can work our way backwards through the network, starting at the output layer and going backwards until we reach the input layer. That way, our approach can be applied to any feedforward neural network as long as the robustness bounds of the different layer types have been established. To be precise, if a given layer computes a function $\boldsymbol{h}$ of its input $\boldsymbol{y}$ and if the following layer has a robustness bound of $\kappa$ in the sense that any adversarial perturbation to this layer has a Euclidean norm of at least $\kappa$, then we want to find a perturbation $\boldsymbol{r}$ such that

$$\|\boldsymbol{h}(\boldsymbol{y} + \boldsymbol{r})\| = \|\boldsymbol{h}(\boldsymbol{y})\| + \kappa.$$

This is clearly a necessary condition for any adversarial perturbation to the given layer. Hence, any adversarial perturbation $\boldsymbol{q}$ to this layer will satisfy $\|\boldsymbol{q}\| \geq \|\boldsymbol{r}\|$. Of course, the output layer of the network will require special treatment. For softmax output layers, $\kappa$ is the norm of the smallest perturbation necessary to change the maximal component of the classification vector.

The obvious downside of this idea is that we most likely introduce cumulative approximation errors which increase as the number of layers of the network increases. In turn, however, we get a flexible and efficient framework which can handle any feedforward architecture composed of known layer types.

### 4.1 Softmax output layers

We now want to find the smallest perturbation $\boldsymbol{r}$ to the input $\boldsymbol{x}$ of a softmax layer such that $f(\boldsymbol{x} + \boldsymbol{r}) \neq f(\boldsymbol{x})$. It can be proven (Theorem A.3) that any such perturbation satisfies

$$\|\boldsymbol{r}\| \geq \min_{c' \neq c} \frac{|(\boldsymbol{w}_{c'} - \boldsymbol{w}_c) \cdot \boldsymbol{x} + b_{c'} - b_c|}{\|\boldsymbol{w}_{c'} - \boldsymbol{w}_c\|},$$

where $f(\boldsymbol{x}) = c$. Moreover, there exist classifiers for which this bound is tight (Theorem A.4).

## 4.2 Fully-connected layers

To analyze the robustness of fully-connected layers to adversarial perturbations, we assume the next layer has a robustness of $\kappa$ (this will usually be the softmax output layer, however there exist CNNs which employ fully-connected layers in other locations than just at the end [Lin et al., 2014]). We then want to find a perturbation $r$ such that

$$\|h_L(x + r)\| = \|h_L(x)\| + \kappa.$$

We find

**Theorem 4.1.** *Let $h_L : \mathbb{R}^d \to \mathbb{R}^n$ be twice differentiable with second-order derivatives bounded by $M$. Then for any $x \in \mathbb{R}^d$,*

$$\|r\| \geq \frac{\sqrt{\|J(x)\|^2 + 2M\sqrt{n}\kappa} - \|J(x)\|}{M\sqrt{n}}, \tag{1}$$

*where $J(x)$ is the Jacobian matrix of $h_L$ at $x$.*

The proof can be found in Appendix A. In Theorem A.5 it is proved that the assumptions on $h_L$ are usually satisfied in practice. The proof of this theorem also yields an efficient algorithm for approximating $M$, a task which otherwise might involve a prohibitively expensive optimization problem.

## 4.3 Convolutional layers

The next layer of the network is assumed to have a robustness bound of $\kappa$, in the sense that any adversarial perturbation $\mathbf{Q}$ to $\mathbf{X}$ must satisfy $\|\mathbf{Q}\|_F \geq \kappa$. We can now attempt to bound the norm of a perturbation $\mathbf{R}$ to $\mathbf{X}$ such that

$$\|\text{ReLU}(\mathbf{W} \star (\mathbf{X} + \mathbf{R}) + b)\|_F = \|\text{ReLU}(\mathbf{W} \star \mathbf{X} + b)\|_F + \kappa.$$

We find

**Theorem 4.2.** *Consider a convolutional layer with filter tensor $\mathbf{W} \in \mathbb{R}^{k \times c \times q \times q}$ and stride $s$ whose input consists of a 3D tensor $\mathbf{X} \in \mathbb{R}^{c \times d \times d}$. Suppose the next layer has a robustness bound of $\kappa$, then any adversarial perturbation to the input of this layer must satisfy*

$$\|\mathbf{R}\|_F \geq \frac{\kappa}{\|\mathbf{W}\|_F}. \tag{2}$$

The proof of Theorem 4.2 can be found in Appendix A.

## 4.4 Pooling layers

To facilitate the analysis of the pooling layers, we make the following assumption which is satisfied by the most common pooling operations (see Appendix B):

**Assumption 4.3.** *The pooling operation satisfies*

$$z_{ijk}(\mathbf{X} + \mathbf{R}) \leq z_{ijk}(\mathbf{X}) + z_{ijk}(\mathbf{R}).$$

We have

**Theorem 4.4.** *Consider a pooling layer whose operation satisfies Assumption 4.3. Let the input be of size $c \times d \times d$ and the receptive field of size $q \times q$. Let the output be of size $c \times t \times t$. If the robustness bound of the next layer is $\kappa$, then the following bounds hold for any adversarial perturbation $\mathbf{R}$:*

- *MAX or average pooling:*

$$\|\mathbf{R}\|_F \geq \frac{\kappa}{t}. \tag{3}$$

- *$L_p$ pooling:*

$$\|\mathbf{R}\|_F \geq \frac{\kappa}{t q^{2/p}}. \tag{4}$$

Proof can be found in Appendix A.

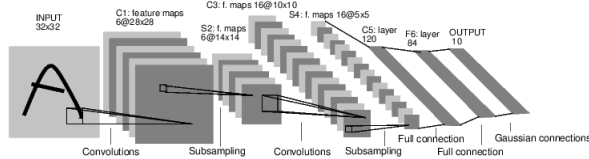

Figure 1: Illustration of LeNet architecture. Image taken from Lecun et al. [1998].

Table 1: Normalized summary of norms of adversarial perturbations found by FGS on MNIST and CIFAR-10 test sets

| Data set | Mean | Median | Std | Min | Max |
|---|---|---|---|---|---|
| MNIST | 0.933448 | 0.884287 | 0.4655439 | 0.000023 | 3.306903 |
| CIFAR-10 | 0.0218984 | 0.0091399 | 0.06103627 | 0.0000012 | 1.6975207 |

## 5 Experimental results

We tested the theoretical bounds on the MNIST and CIFAR-10 test sets using the Caffe [Jia et al., 2014] implementation of LeNet [Lecun et al., 1998]. The MNIST data set [LeCun et al., 1998] consists of 70,000 $28 \times 28$ images of handwritten digits; the CIFAR-10 data set [Krizhevsky and Hinton, 2009] consists of 60,000 $32 \times 32$ RGB images of various natural scenes, each belonging to one of ten possible classes. The architecture of LeNet is depicted in Figure 1. The kernels of the two convolutional layers will be written as $\mathbf{W}_1$ and $\mathbf{W}_2$, respectively. The output sizes of the two pooling layers will be written as $t_1$ and $t_2$. The function computed by the first fully-connected layer will be denoted by $h$ with Jacobian $J$. The last fully-connected layer has a weight matrix $V$ and bias vector $b$. For an input sample $x$, the theoretical lower bound on the adversarial robustness of the network with respect to $x$ is given by $\kappa_1$, where

$$\kappa_6 = \min_{c' \neq c} \frac{|(\boldsymbol{v}_{c'} - \boldsymbol{v}_c) \cdot \boldsymbol{x} + b_{c'} - b_c|}{\|\boldsymbol{v}_{c'} - \boldsymbol{v}_c\|}, \qquad \kappa_5 = \frac{\sqrt{\|\boldsymbol{J}(\boldsymbol{x})\|^2 + 2M\sqrt{500}\kappa_6} - \|\boldsymbol{J}(\boldsymbol{x})\|}{M\sqrt{500}},$$

$$\kappa_4 = \frac{\kappa_5}{t_2}, \qquad\qquad \kappa_3 = \frac{\kappa_4}{\|\mathbf{W}_2\|_F},$$

$$\kappa_2 = \frac{\kappa_3}{t_1}, \qquad\qquad \kappa_1 = \frac{\kappa_2}{\|\mathbf{W}_1\|_F}.$$

Because our method only computes norms and does not provide a way to generate actual adversarial perturbations, we used the fast gradient sign method (FGS) [Goodfellow et al., 2015] to adversarially perturb each sample in the test sets in order to assess the tightness of our theoretical bounds. FGS linearizes the cost function of the network to obtain an estimated perturbation

$$\boldsymbol{\eta} = \varepsilon \text{sign} \nabla_{\boldsymbol{x}} \mathcal{L}(\boldsymbol{x}, \theta).$$

Here, $\varepsilon > 0$ is a parameter of the algorithm, $\mathcal{L}$ is the loss function and $\theta$ is the set of parameters of the network. The magnitudes of the perturbations found by FGS depend on the choice of $\varepsilon$, so we had to minimize this value in order to obtain the smallest perturbations the FGS method could supply. This was accomplished using a simple binary search for the smallest value of $\varepsilon$ which still resulted in misclassification. As the MNIST and CIFAR-10 samples have pixel values within the range $[0, 255]$, we upper-bounded $\varepsilon$ by 100.

No violations of the bounds were detected in our experiments. Figure 2 shows histograms of the norms of adversarial perturbations found by FGS and Table 1 summarizes their statistics. Histograms of the theoretical bounds of all samples in the test set are shown in Figure 3; their statistics are summarized in Table 2. Note that the statistics of Tables 1 and 2 have been normalized by dividing them by the dimensionality of their respective data sets (i.e. $28 \times 28$ for MNIST and $3 \times 32 \times 32$ for CIFAR-10) to allow for a meaningful comparison between the two networks. Figure 4 provides histograms of the per-sample log-ratio between the norms of the adversarial perturbations and their corresponding theoretical lower bounds.

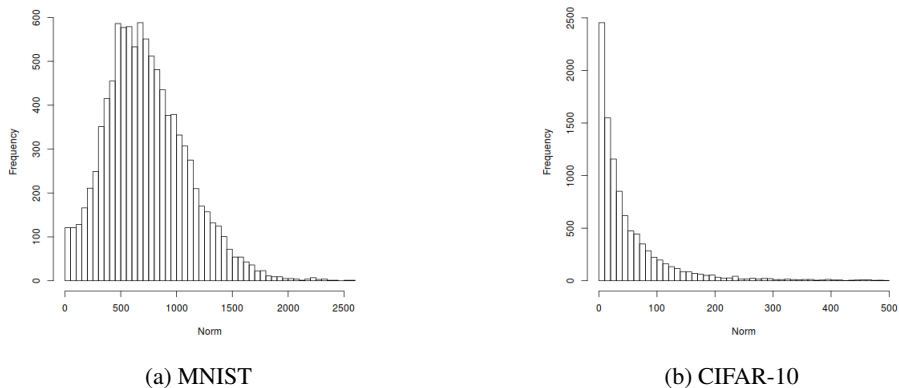

(a) MNIST          (b) CIFAR-10

Figure 2: Histograms of norms of adversarial perturbations found by FGS on MNIST and CIFAR-10 test sets

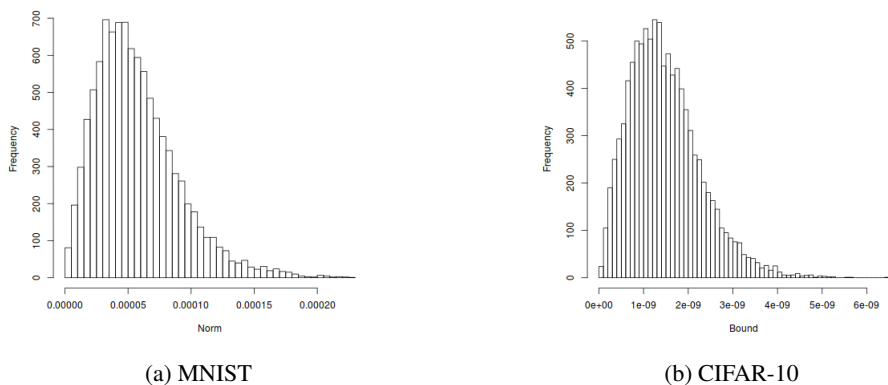

(a) MNIST          (b) CIFAR-10

Figure 3: Histograms of theoretical bounds on MNIST and CIFAR-10 test sets

Although the theoretical bounds on average deviate considerably from the perturbations found by FGS, one has to take into consideration that the theoretical bounds were constructed to provide a worst-case estimate for the norms of adversarial perturbations. These estimates may not hold for all (or even most) input samples. Furthermore, the smallest perturbations we were able to generate on the two data sets have norms that are much closer to the theoretical bound than their averages (0.0179 for MNIST and 0.0000012 for CIFAR-10). This indicates that the theoretical bound is not necessarily very loose, but rather that very small adversarial perturbations occur with non-zero probability on natural samples. Note also that the FGS method does not necessarily generate minimal perturbations even with the smallest choice of $\varepsilon$: the method depends on the linearity hypothesis and uses a first-order Taylor approximation of the loss function. Higher-order methods may find much smaller perturbations by exploiting non-linearities in the network, but these are generally much less efficient than FGS.

There is a striking difference in magnitude between MNIST and CIFAR-10 of both the empirical and theoretical perturbations: the perturbations on MNIST are much larger than the ones found for

Table 2: Normalized summary of theoretical bounds on MNIST and CIFAR-10 test sets

| Data set | Mean | Median | Std | Min | Max |
|----------|------|--------|-----|-----|-----|
| MNIST | 7.274e−8 | 6.547e−8 | 4.229566e−8 | 4.073e−10 | 2.932e−7 |
| CIFAR-10 | 4.812e−13 | 4.445e−13 | 2.605381e−13 | 7.563e−15 | 2.098e−12 |

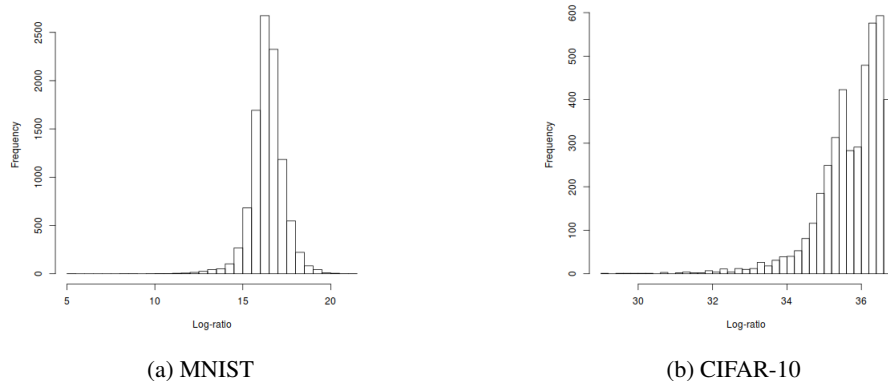

|              |              |
|:------------:|:------------:|
| (a) MNIST    | (b) CIFAR-10 |

Figure 4: Histograms of the per-sample log-ratio between adversarial perturbation and lower bound for MNIST and CIFAR-10 test sets. A higher ratio indicates a bigger deviation of the theoretical bound from the empirical norm.

CIFAR-10. This result can be explained by the linearity hypothesis of Goodfellow et al. [2015]. The input samples of CIFAR-10 are much larger in dimensionality than MNIST samples, so the linearity hypothesis correctly predicts that networks trained on CIFAR-10 are more susceptible to adversarial perturbations due to the highly linear behavior these classifiers are conjectured to exhibit. However, these differences may also be related to the fact that LeNet achieves much lower accuracy on the CIFAR-10 data set than it does on MNIST (over 99% on MNIST compared to about 60% on CIFAR-10).

## 6 Conclusion and future work

Despite attracting a significant amount of research interest, a precise characterization of adversarial examples remains elusive. In this paper, we derived lower bounds on the norms of adversarial perturbations in terms of the model parameters of feedforward neural network classifiers consisting of convolutional layers, pooling layers, fully-connected layers and softmax layers. The bounds can be computed efficiently and thus may serve as an aid in model selection or the development of methods to increase the robustness of classifiers. They enable one to assess the robustness of a classifier without running extensive tests, so they can be used to compare different models and quickly select the one with highest robustness. Furthermore, the bounds enjoy a theoretical guarantee that no adversarial perturbation could ever be smaller, so methods which increase these bounds may make classifiers more robust. We tested the validity of our bounds on MNIST and CIFAR-10 and found no violations. Comparisons with adversarial perturbations generated using the fast gradient sign method suggest that these bounds can be close to the actual norms in the worst case.

We have only derived lower bounds for feedforward networks consisting of fully-connected layers, convolutional layers and pooling layers. Extending this analysis to recurrent networks and other types of layers such as Batch Normalization [Ioffe and Szegedy, 2015] and Local Response Normalization [Krizhevsky et al., 2012] is an obvious avenue for future work.

It would also be interesting to quantify just how tight the above bounds really are. In the absence of a precise characterization of adversarial examples, the only way to do this would be to generate adversarial perturbations using optimization techniques that make no assumptions on their underlying cause. Szegedy et al. [2014] use a box-constrained L-BFGS approach to generate adversarial examples without any assumptions, so using this method for comparison could provide a more accurate picture of how tight the theoretical bounds are. It is much less efficient than the FGS method, however.

The analysis presented here is a "modular" one: we consider each layer in isolation, and derive bounds on their robustness in terms of the robustness of the next layer. However, it may also be insightful to study the relationship between the number of layers, the breadth of each layer and the robustness of the network. Providing estimates on the approximation errors incurred by this layer-wise approach could also be useful.

Finally, there is currently no known precise characterization of the trade-off between classifier robustness and accuracy. Intuitively, one might expect that as the robustness of the classifier increases, its accuracy will also increase up to a point since it is becoming more robust to adversarial perturbations. Once the robustness exceeds a certain threshold, however, we expect the accuracy to drop because the decision surfaces are becoming too flat and the classifier becomes too insensitive to changes. Having a precise characterization of this relationship between robustness and accuracy may aid methods designed to protect classifiers against adversarial examples while also maintaining state-of-the-art accuracy.

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
