[Supplementary Material]

# A  Proofs of lower bounds

## A.1  Softmax output layers

Let $f$ be a linear classifier. For an observation $x$ and any class $c \neq f(x)$, we can compute the norm of the smallest perturbation $r$ such that $f(x + r) = c$. Let us denote this perturbation by $\Delta_{\text{adv}}(x; f, c)$:

$$\Delta_{\text{adv}}(x; f, c) = \min_{r \in \mathbb{R}^d} \{\|r\| \mid f(x + r) = c\}. \tag{5}$$

It is easily seen that

$$\Delta_{\text{adv}}(x; f) = \min_{c \neq f(x)} \{\Delta_{\text{adv}}(x; f, c)\}. \tag{6}$$

The restriction that $c \neq f(x)$ is important, since otherwise we would get a degenerate solution of $\Delta_{\text{adv}}(x; f) = 0$ for all $x$. Intuitively, $\Delta_{\text{adv}}(x; f)$ is the norm of the projection of $x$ onto the class "closest" to $x$ but distinct from $f(x)$. A necessary condition for any adversarial perturbation $q$ is given by the following

**Lemma A.1.** *Let $f$ be a linear classifier and let $q$ be an adversarial perturbation for an instance $x$ where $f(x) = c$ and $f(x + q) = c' \neq c$. There exists an $\alpha \in (0, 1)$ such that $w_c \cdot (x + \alpha q) + b_c = w_{c'} \cdot (x + \alpha q) + b_{c'}$.*

*Proof.* A little algebra shows

$$\alpha = \frac{(w_{c'} - w_c) \cdot x + b_{c'} - b_c}{(w_c - w_{c'}) \cdot q}. \tag{7}$$

It remains to be proven that $0 < \alpha < 1$. Note that, by assumption, $w_c \cdot x + b_c > w_{c'} \cdot x + b_{c'}$ and $w_c \cdot (x + q) + b_c < w_{c'} \cdot (x + q) + b_{c'}$. This implies $(w_{c'} - w_c) \cdot x + b_{c'} - b_c < 0$ and $(w_c - w_{c'}) \cdot q < 0$, so $\alpha > 0$. Furthermore, since $w_c \cdot (x + q) + b_c < w_{c'} \cdot (x + q) + b_{c'}$, we find $\alpha < 1$. □

In other words, Lemma A.1 states that for any linear classifier $f$, if we want to adversarially perturb an input $x$ so its assigned label changes from $c$ to $c'$, we must cross the boundary where $f$ assigns equal probability to those two classes. Note that this condition is only necessary, not sufficient: it is perfectly possible to cross this boundary for the classes $c$ and $c'$ at a point where some other class $c''$ is still more likely than either $c$ or $c'$, but we must cross this boundary nonetheless. Using Lemma A.1 we find

**Lemma A.2.** *Let $f$ be a linear classifier and let $x$ be any input such that $f(x) = c'$. Then for all classes $c \neq c'$,*

$$\Delta_{\text{adv}}(x; f, c) \geq \frac{|(w_{c'} - w_c) \cdot x + b_{c'} - b_c|}{\|w_{c'} - w_c\|}.$$

*Proof.* We can find the norm of the smallest perturbation $r$ such that $w_c \cdot (x + r) + b_c = w_{c'} \cdot (x + r) + b_{c'}$ using the following Lagrangian:

$$\mathcal{L} = \|r\|^2 + \lambda((w_{c'} - w_c) \cdot (x + r) + b_{c'} - b_c)$$

The solution to this optimization problem is given by

$$r = \frac{(w_{c'} - w_c) \cdot x + b_{c'} - b_c}{\|w_{c'} - w_c\|^2}(w_c - w_{c'}).$$

Taking norms we get

$$\|r\| = \frac{|(w_{c'} - w_c) \cdot x + b_{c'} - b_c|}{\|w_{c'} - w_c\|}.$$

By Lemma A.1 and the construction of $r$, any adversarial perturbation $q$ must satisfy $\|q\| > \|r\|$. In particular, any adversarial perturbation $q$ such that $f(x + q) = c$ satisfies $\|q\| > \|r\|$. Hence $\Delta_{\text{adv}}(x; f, c) \geq \|r\|$. □

The following result is a trivial consequence of Lemma A.2:

**Theorem A.3.** *Let $f$ be a linear classifier. Then for all inputs $\boldsymbol{x}$ where $f(\boldsymbol{x}) = c$,*

$$\Delta_{\mathrm{adv}}(\boldsymbol{x}; f) \geq \min_{c' \neq c} \frac{|(\boldsymbol{w}_{c'} - \boldsymbol{w}_c) \cdot \boldsymbol{x} + b_{c'} - b_c|}{\|\boldsymbol{w}_{c'} - \boldsymbol{w}_c\|}.$$

Note how Theorem A.3 confirms the intuition that the smallest adversarial perturbation to an instance $\boldsymbol{x}$ is bounded from below by the orthogonal projection of $\boldsymbol{x}$ onto the class closest to $\boldsymbol{x}$ but distinct from $f(\boldsymbol{x})$, since this is exactly the quantity on the right-hand side of the inequality.

**Theorem A.4.** *The bound of Theorem A.3 is tight.*

*Proof.* Let $f$ be a classifier for $C = d \geq 2$ classes where

$$f(\boldsymbol{x}) = \begin{cases} 1 & \text{if } x_1 > x_2, \ldots, x_C \\ 2 & \text{if } x_2 > x_1, x_3, \ldots, x_C \\ \vdots \\ C & \text{if } x_C > x_1, \ldots, x_{C-1} \end{cases}.$$

This classifier is linear since it can be characterized by linear functions of the form

$$f_i(\boldsymbol{x}) = \boldsymbol{e}_i \cdot \boldsymbol{x} = x_i.$$

Hence, it is subject to the bound of Theorem A.3, which in this case simplifies to

$$\Delta_{\mathrm{adv}}(\boldsymbol{x}; f) \geq \min_{c' \neq c} |x_{c'} - x_c|.$$

It is easily seen, however, that this bound is exact for this particular classifier, i.e.

$$\Delta_{\mathrm{adv}}(\boldsymbol{x}; f) = \min_{c' \neq c} |x_{c'} - x_c|.$$

The reason being that $f$ classifies an input $\boldsymbol{x}$ into the class corresponding to the index of the maximal component of $\boldsymbol{x}$. Thus, $f(\boldsymbol{x}) = c$ if and only if $x_c$ is the maximal component of $\boldsymbol{x}$. To find the norm of the minimal perturbation $\boldsymbol{r}$ such that $f(\boldsymbol{x} + \boldsymbol{r}) \neq c$, one simply takes $\boldsymbol{r} = (x_c - x_{c'})\boldsymbol{e}_{c'}$ where $c'$ is the index of the second-highest component of $\boldsymbol{x}$. Clearly $f(\boldsymbol{x} + \boldsymbol{r}) = c' \neq c$ and $\|\boldsymbol{r}\| = |x_{c'} - x_c|$ is minimal. $\square$

## A.2 Fully-connected layers

Applying Taylor's theorem to $\boldsymbol{h}_L$ we obtain

$$\boldsymbol{h}_L(\boldsymbol{x} + \boldsymbol{r}) = \boldsymbol{h}_L(\boldsymbol{x}) + \boldsymbol{J}(\boldsymbol{x})\boldsymbol{r} + \boldsymbol{\varepsilon},$$

where

$$\|\boldsymbol{\varepsilon}\| \leq \frac{M}{2}\sqrt{n}\,\|\boldsymbol{r}\|^2.$$

Here, $M$ is a real number bounding the absolute value of all second-order derivatives of $\boldsymbol{h}_L$ from above. Thus $\boldsymbol{q} = \boldsymbol{J}(\boldsymbol{x})\boldsymbol{r} + \boldsymbol{\varepsilon}$ and

$$\|\boldsymbol{q}\| \leq \|\boldsymbol{J}(\boldsymbol{x})\|\,\|\boldsymbol{r}\| + \frac{M}{2}\sqrt{n}\,\|\boldsymbol{r}\|^2.$$

This is a quadratic inequality in $\|\boldsymbol{r}\|$ whose solution is given by

$$\|\boldsymbol{r}\| \geq \frac{\sqrt{\|\boldsymbol{J}(\boldsymbol{x})\|^2 + 2M\sqrt{n}\,\|\boldsymbol{q}\|} - \|\boldsymbol{J}(\boldsymbol{x})\|}{M\sqrt{n}}.$$

By Theorem A.3 we know

$$\|\boldsymbol{q}\| \geq \min_{c' \neq c} \frac{|(\boldsymbol{w}_c - \boldsymbol{w}_{c'}) \cdot \boldsymbol{h}(\boldsymbol{x})|}{\|\boldsymbol{w}_c - \boldsymbol{w}_{c'}\|}.$$

Hence the theorem follows.

One might rightly ask how realistic this result actually is. After all, we needed to assume that the function $\boldsymbol{h}_L$ was twice differentiable and had bounded second-order derivatives. In this section, we will try to show that these assumptions are quite realistic by deriving them from other realistic assumptions on the activation functions. Specifically, we have the following

**Theorem A.5.** *Consider fully-connected layers $\boldsymbol{h}_1, \ldots, \boldsymbol{h}_L$ whose activation functions $g_i$ are twice differentiable on $\mathbb{R}$. Assume also that for each $g_i$ there exist $N_i > 0$ and $M_i > 0$ such that $|g_i'(x)| \leq N_i$ and $|g_i''(x)| \leq M_i$ for all $x$. Then $\boldsymbol{h}_L(\boldsymbol{x}) = [h_L^{(1)}(\boldsymbol{x}), \ldots, h_L^{(n)}(\boldsymbol{x})]^T$ satisfies the following properties:*

1. *each $h_L^{(i)}$ is twice differentiable;*

2. *$|\partial^\alpha h_L^{(i)}(\boldsymbol{x})| \leq M$ for all $i$, $|\alpha| = 2$ and some $M > 0$;*

3. *$|\partial^\alpha h_L^{(i)}(\boldsymbol{x})| \leq N$ for all $i$, $|\alpha| = 1$ and some $N > 0$.*

*Proof.* The proof proceeds by induction on $L$. The case where $L = 0$ is trivial, so suppose

1. each $h_L^{(i)}$ is twice differentiable;

2. $|\partial^\alpha h_L^{(i)}(\boldsymbol{x})| \leq M'$ for all $i$, $|\alpha| = 2$ and some $M' > 0$;

3. $|\partial^\alpha h_L^{(i)}(\boldsymbol{x})| \leq N'$ for all $i$, $|\alpha| = 1$ and some $N' > 0$.

We need to show that $\boldsymbol{h}_{L+1} : \mathbb{R}^d \to \mathbb{R}^n : \boldsymbol{x} \mapsto [h_{L+1}^{(1)}(\boldsymbol{x}), \ldots, h_{L+1}^{(n)}(\boldsymbol{x})]^T$ then satisfies the following properties:

1. $h_{L+1}^{(i)}$ is twice differentiable for all $i$;

2. there exists an $M > 0$ such that $|\partial^\alpha h_{L+1}^{(i)}(\boldsymbol{x})| \leq M$ for all $\boldsymbol{x}$, $|\alpha| = 2$ and $i$;

3. there exists an $N > 0$ such that $|\partial^\alpha h_{L+1}^{(i)}(\boldsymbol{x})| \leq N$ for all $\boldsymbol{x}$, $|\alpha| = 1$ and $i$.

Since $\boldsymbol{h}_{L+1}(\boldsymbol{x}) = g_{L+1}(\boldsymbol{V}_{L+1}\boldsymbol{h}_L(\boldsymbol{x}) + \boldsymbol{b}_{L+1})$ we find

$$h_{L+1}^{(i)}(\boldsymbol{x}) = g_{L+1}\left(\sum_j v_{L+1,i,j}h_L^{(j)}(\boldsymbol{x}) + b_{L+1,i}\right).$$

Clearly, since $g_{L+1}$ is twice differentiable by assumption and $h_L^{(j)}$ is twice differentiable for all $j$ by the induction hypothesis, $h_{L+1}^{(i)}$ is twice differentiable for all $i$. This shows Item 1. To show Item 2, we distinguish two cases. First, let

$$\alpha_j = \begin{cases} 2 & j = k \\ 0 & \text{otherwise} \end{cases}$$

for some $k \in \{1, \ldots, d\}$. Then

$$\partial^\alpha h_{L+1}^{(i)} = \frac{\partial^2 h_{L+1}^{(i)}}{\partial x_k^2} = \frac{\partial}{\partial x_k}\left(g_{L+1}'\left(\sum_j v_{L+1,i,j}h_L^{(j)}(\boldsymbol{x}) + b_{L+1,i}\right)\sum_j v_{L+1,i,j}\frac{\partial}{\partial x_k}h_L^{(j)}(\boldsymbol{x})\right)$$

$$= g_{L+1}''\left(\sum_j v_{L+1,i,j}h_L^{(j)}(\boldsymbol{x}) + b_{L+1,i}\right)\left(\sum_j v_{L+1,i,j}\frac{\partial}{\partial x_k}h_L^{(j)}(\boldsymbol{x})\right)^2 +$$

$$g_{L+1}'\left(\sum_j v_{L+1,i,j}h_L^{(j)}(\boldsymbol{x}) + b_{L+1,i}\right)\sum_j v_{L+1,i,j}\frac{\partial^2}{\partial x_k^2}h_L^{(j)}(\boldsymbol{x}).$$

Taking absolute values and applying the induction hypothesis, this yields

$$|\partial^\alpha h_{L+1}^{(i)}| \leq M_{L+1}\left(N'\sum_j |v_{L+1,i,j}|\right)^2 + N_{L+1}M'\sum_j |v_{L+1,i,j}|.$$

Hence we may choose

$$M = M_{L+1}\left(N's\right)^2 + N_{L+1}M's,$$

where

$$s = \max_i \sum_j |v_{L+1,i,j}|.$$

For the second case, let

$$\alpha_j = \begin{cases} 1 & j \in \{k_1, k_2\} \\ 0 & \text{otherwise} \end{cases}$$

for $k_1, k_2 \in \{1, \ldots, d\}$ and $k_1 < k_2$. Of course, this case only applies when $d > 1$. We find

$$\partial^\alpha h_{L+1}^{(i)} = \frac{\partial^2 h_{L+1}^{(i)}}{\partial x_{k_1} \partial x_{k_2}} = \frac{\partial}{\partial x_{k_2}}\left(g'_{L+1}\left(\sum_j v_{L+1,i,j} h_L^{(j)}(\boldsymbol{x}) + b_{L+1,i}\right) \sum_j v_{L+1,i,j} \frac{\partial}{\partial x_{k_1}} h_L^{(j)}(\boldsymbol{x})\right)$$

$$= g''_{L+1}\left(\sum_j v_{L+1,i,j} h_L^{(j)}(\boldsymbol{x}) + b_{L+1,i}\right) \sum_j v_{L+1,i,j} \frac{\partial}{\partial x_{k_2}} h_L^{(j)}(\boldsymbol{x}) \sum_j v_{L+1,i,j} \frac{\partial}{\partial x_{k_1}} h_L^{(j)}(\boldsymbol{x})$$

$$+ g'_{L+1}\left(\sum_j v_{L+1,i,j} h_L^{(j)}(\boldsymbol{x}) + b_{L+1,i}\right) \sum_j v_{L+1,i,j} \frac{\partial^2}{\partial x_{k_1} \partial x_{k_2}} h_L^{(j)}(\boldsymbol{x})$$

Again, taking absolute values and applying the induction hypothesis:

$$|\partial^\alpha h_{L+1}^{(i)}(\boldsymbol{x})| \leq M_{L+1}\left(N' \sum_j |v_{L+1,i,j}|\right)^2 + N_{L+1}M' \sum_j |v_{L+1,i,j}|$$

The result is identical to the first case.

Finally, to show Item 3, we let

$$\alpha_j = \begin{cases} 1 & j = k \\ 0 & \text{otherwise} \end{cases}$$

for some $k \in \{1, \ldots, d\}$. Then

$$\partial^\alpha h_{L+1}^{(i)} = \frac{\partial h_{L+1}^{(i)}}{\partial x_k} = g'_{L+1}\left(\sum_j v_{L+1,i,j} h_L^{(j)}(\boldsymbol{x}) + b_{L+1,i}\right) \sum_j v_{L+1,i,j} \frac{\partial}{\partial x_k} h_L^{(j)}(\boldsymbol{x}).$$

Again taking absolute values and applying the induction hypothesis, we find

$$|\partial^\alpha h_{L+1}^{(i)}| \leq N_{L+1}N' \sum_j |v_{L+1,i,j}|.$$

Hence we may choose

$$N = (N_{L+1}N')\max_i \sum_j |v_{L+1,i,j}|.$$

This completes the proof. $\qquad\square$

By Theorem A.5, in order for Theorem 4.1 to hold it is sufficient that the activation functions of the MLP in question be twice differentiable and have bounded first and second derivatives. These assumptions are not unrealistic: they are satisfied by the logistic sigmoid function, for example. The logistic sigmoid is in fact just a scaled and shifted version of the hyperbolic tangent:

$$\text{sigm}(x) = \frac{1}{2}\tanh\left(\frac{x}{2}\right) + \frac{1}{2}. \tag{8}$$

Since $\tanh$ is twice differentiable, so is $\mathrm{sigm}$. Moreover, the first and second derivatives of $\tanh$ are bounded by 1, so the first and second derivatives of $\mathrm{sigm}$ are bounded as well. In fact, $|\tanh(x)| \leq 1$ for all $x$.

The ReLU activation function presents some problems, however, as it is not differentiable at zero. Gradient-based optimization requires all activation functions be differentiable, though, so in practice either a smooth approximation to ReLU is used which is differentiable, such as the softplus function $\ln(1+\exp(x))$, or the value of the derivative is simply set to zero at the origin (Glorot et al. [2011]). In both cases it can be seen that ReLU (as it is used in practice) also satisfies the necessary assumptions for Theorem 4.1 to hold.

Note also how the proof of Theorem A.5 yields an efficient algorithm for approximating the $M$ parameter used in the lower bound. Algorithm A.1 shows how this can be done in $\mathcal{O}(n)$ time where $n$ is the number of parameters of the neural network.

---

**Algorithm A.1:** Computation of $M$

**Data:** MLP $f$ with $L$ hidden layers, activation functions $g_i$ satisfying $|g_i'(x)| \leq A_i$ and $|g_i''(x)| \leq B_i$ for all $x$.
**Result:** a value $M$ satisfying $|\partial^\alpha h_L^{(i)}(\boldsymbol{x})| \leq M$ for all $\boldsymbol{x}$, $|\alpha| = 2$ and $i$
**begin**
    $M_0 \leftarrow 0$
    $N_0 \leftarrow 1$
    **for** $i$ *from 1 to $L$* **do**
        $s \leftarrow \max_j \sum_k |v_{i,j,k}|$
        $M_i \leftarrow B_i N_{i-1}^2 s^2 + A_i M_{i-1} s$
        $N_i \leftarrow A_i N_{i-1} s$
    **end**
    **return** $M_L$
**end**

---

### A.3 Convolutional layers

Using Lemma B.1 we find[1]

$$\begin{aligned}
\|\mathrm{ReLU}(\mathbf{W} \star (\mathbf{X} + \mathbf{R}) + b)\|_F &= \|\mathrm{ReLU}(\mathbf{W} \star \mathbf{X} + \mathbf{W} \star \mathbf{R} + b)\|_F \\
&\leq \|\mathrm{ReLU}(\mathbf{W} \star \mathbf{X} + b) + \mathrm{ReLU}(\mathbf{W} \star \mathbf{R})\|_F \\
&\leq \|\mathrm{ReLU}(\mathbf{W} \star \mathbf{X} + b)\|_F + \|\mathrm{ReLU}(\mathbf{W} \star \mathbf{R})\|_F .
\end{aligned}$$

This yields

$$\|\mathrm{ReLU}(\mathbf{W} \star \mathbf{R})\|_F \geq \kappa.$$

A necessary condition for this equality to hold is (Lemma B.1)

$$\|\mathbf{W} \star \mathbf{R}\|_F = \kappa. \tag{9}$$

Thanks to Lemma B.3, we may rewrite Equation (9) as

$$\|\mathbf{R}\|_F \geq \frac{\kappa}{\|\mathbf{W}\|_F}.$$

### A.4 Pooling layers

Assuming any adversarial perturbation $\mathbf{Q}$ to the input of the next layer needs to satisfy $\|\mathbf{Q}\|_F \geq \kappa$, Assumption 4.3 implies we have to solve the following equation:

$$\|\mathbf{Z}(\mathbf{R})\|_F \geq \kappa. \tag{10}$$

A necessary condition for Equation (10) to hold is to have

$$|z_{ijk}(\mathbf{R})| \geq \frac{\kappa}{t} \tag{11}$$

for at least one element $z_{ijk}(\mathbf{R})$. How this can be done depends on the precise nature of the pooling operation.

### A.4.1 MAX-pooling

MAX-pooling reduces the dimensionality of the input by taking the maximum of all $q \times q$ regions within the receptive field:

$$z_{ijk}(\mathbf{X}) = \max\{x_{inm} \mid (n,m) \in I(j,k)\}.$$

Proof that MAX-pooling satisfies Assumption 4.3 is given in Lemma B.4. In order to satisfy Equation (11), it is clearly necessary to set at least one component of $\mathbf{R}$ equal to or greater than $\kappa/t$ in absolute value. This yields

$$\|\mathbf{R}\|_F \geq \frac{\kappa}{t}. \tag{12}$$

### A.4.2 $L_p$ pooling

An $L_p$ pooling layer produces as output the $L_p$ norm of its input:

$$z_{ijk}(\mathbf{X}) = \left( \sum_{(n,m) \in I} |x_{inm}|^p \right)^{\frac{1}{p}}.$$

We will write $\boldsymbol{v}_{ijk}$ for the vector whose $L_p$ norm is taken in the computation of $z_{ijk}$. Proof that $L_p$-pooling satisfies Assumption 4.3 is given in Lemma B.5. In order to satisfy Equation (11), we must have

$$\|\boldsymbol{v}_{ijk}(\mathbf{R})\|_p \geq \frac{\kappa}{t} \tag{13}$$

for some $i, j, k$. For Equation (13) to be satisfied, there must be at least one element $r_{lmn}$ in some receptive field of $\mathbf{R}$ such that

$$|r_{lmn}| \geq \frac{\kappa}{t q^{2/p}}.$$

Since there will be at least one receptive field, at least one element of $\mathbf{R}$ must satisfy this requirement and hence

$$\|\mathbf{R}\|_F \geq \frac{\kappa}{t q^{2/p}}. \tag{14}$$

Note the nice property that as $p \to \infty$ we find

$$\|\mathbf{R}\|_F \geq \frac{\kappa}{t},$$

which is the bound for MAX-pooling.

### A.4.3 Average pooling

An average pooling layer takes the average of all its inputs:

$$z_{ijk}(\mathbf{X}) = \frac{1}{q^2} \sum_{(n,m) \in I} x_{inm}.$$

Proof that average pooling satisfies Assumption 4.3 is given in Lemma B.6. Note that, contrary to all the other pooling operations studied here, Assumption 4.3 holds with equality in the case of average pooling. To satisfy Equation (11), it is necessary that at least one element of $\mathbf{R}$ be greater than or equal to $\kappa/t$ in absolute value. We thus find

$$\|\mathbf{R}\|_F \geq \frac{\kappa}{t}. \tag{15}$$

# B Auxiliary results

**Lemma B.1.**

1. *Let $a, b \in \mathbb{R}$, then*
$$\mathrm{ReLU}(a + b) \leq \mathrm{ReLU}(a) + \mathrm{ReLU}(b).$$

2. *Let $\mathbf{W}, \mathbf{X}, \mathbf{R}$ be tensors and $s \in \mathbb{N}$, then*
$$\mathbf{W} \star_s (\mathbf{X} + \mathbf{R}) = \mathbf{W} \star_s \mathbf{X} + \mathbf{W} \star_s \mathbf{R}.$$

3. *Let $\mathbf{X}$ be any real-valued tensor, then*
$$\|\mathrm{ReLU}(\mathbf{X})\|_F \leq \|\mathbf{X}\|_F.$$

*Proof.*

1. We distinguish four cases:
   - $a, b > 0$:
   $$\mathrm{ReLU}(a + b) = a + b = \mathrm{ReLU}(a) + \mathrm{ReLU}(b).$$
   - $a > 0$ and $b \leq 0$:
   $$\mathrm{ReLU}(a + b) \leq a + b \leq a = \mathrm{ReLU}(a) + \mathrm{ReLU}(b).$$
   - $a \leq 0$ and $b > 0$:
   $$\mathrm{ReLU}(a + b) \leq a + b \leq b = \mathrm{ReLU}(a) + \mathrm{ReLU}(b).$$
   - $a, b < 0$:
   $$\mathrm{ReLU}(a + b) = 0 = \mathrm{ReLU}(a) + \mathrm{ReLU}(b).$$

2. Suppose $\mathbf{W} \in \mathbb{R}^{n_1 \times \cdots \times n_d}$ and $\mathbf{X}, \mathbf{R} \in \mathbb{R}^{m_1 \times \cdots m_d}$, then
$$
\begin{aligned}
(\mathbf{W} \star_s (\mathbf{X} + \mathbf{R}))_{i_1 \ldots i_d} &= \sum_{j_1, \ldots, j_d} w_{j_1 \ldots j_d}\left(x_{i_1 + s(j_1 - 1), \ldots, i_d + s(j_d - 1)} + r_{i_1 + s(j_1 - 1), \ldots, i_d + s(j_d - 1)}\right) \\
&= \sum_{j_1, \ldots, j_d} w_{j_1 \ldots j_d} x_{i_1 + s(j_1 - 1), \ldots, i_d + s(j_d - 1)} + \sum_{j_1, \ldots, j_d} w_{j_1 \ldots j_d} r_{i_1 + s(j_1 - 1), \ldots, i_d + s(j_d - 1)} \\
&= (\mathbf{W} \star_s \mathbf{X})_{i_1 \ldots i_d} + (\mathbf{W} \star_s \mathbf{R})_{i_1 \ldots i_d}.
\end{aligned}
$$

3. Let $\mathbf{X} \in \mathbb{R}^{n_1 \times \cdots \times n_d}$. We compute:
$$\|\mathrm{ReLU}(\mathbf{X})\|_F^2 = \sum_{i_1, \ldots, i_d} \max\{0, x_{i_1 \ldots i_d}\}^2 \leq \sum_{i_1, \ldots, i_d} x_{i_1 \ldots i_d}^2 = \|\mathbf{X}\|_F^2.$$

$\square$

**Lemma B.2.** *Let $a_i, b_i$ be non-negative real numbers for $i = 1, \ldots, n$. Then*
$$\sum_i a_i b_i \leq \left(\sum_i a_i\right)\left(\sum_i b_i\right).$$

*Proof.* This is easy to see by direct computation:
$$\left(\sum_i a_i\right)\left(\sum_i b_i\right) = \sum_i a_i \left(\sum_j b_j\right).$$

Since all terms are non-negative, we have
$$b_i \leq \sum_j b_j$$

for all $i$. Hence the result follows.

$\square$

**Lemma B.3.** *Let* $\mathbf{W}$ *and* $\mathbf{R}$ *be as above, then*

$$\|\mathbf{W} \star \mathbf{R}\|_F \le \|\mathbf{W}\|_F \|\mathbf{R}\|_F.$$

*Proof.* We compute:

$$\|\mathbf{W} \star \mathbf{R}\|_F^2 = \sum_i \|\mathbf{W}_i \star \mathbf{R}\|_F^2 = \sum_{i,j,k} (\mathbf{W}_i \star \mathbf{R})_{jk}^2$$

$$= \sum_{i,j,k,l,m,n} w_{ilmn}^2 r_{l,m+s(j-1),n+s(k-1)}^2.$$

Using Lemma B.2:

$$\sum w_{ilmn}^2 r_{l,m+s(j-1),n+s(k-1)}^2 \le \left(\sum w_{ilmn}^2\right) \left(\sum r_{l,m+s(j-1),n+s(k-1)}^2\right)$$

$$\le \left(\sum w_{ilmn}^2\right) \left(\sum r_{lmn}^2\right)$$

$$= \|\mathbf{W}\|_F^2 \|\mathbf{R}\|_F^2.$$

We can thus conclude

$$\|\mathbf{W} \star \mathbf{R}\|_F \le \|\mathbf{W}\|_F \|\mathbf{R}\|_F.$$

$\square$

**Lemma B.4.** *MAX-pooling satisfies Assumption 4.3.*

*Proof.* We compute:

$$z_{ijk}(\mathbf{X} + \mathbf{R}) = \max\{x_{inm} + r_{inm} \mid (n,m) \in I(j,k)\}$$
$$\le \max\{x_{inm} \mid (n,m) \in I(j,k)\} + \max\{r_{inm} \mid (n,m) \in I(j,k)\}$$
$$= z_{ijk}(\mathbf{X}) + z_{ijk}(\mathbf{R}).$$

$\square$

**Lemma B.5.** $L_p$ *pooling satisfies Assumption 4.3.*

*Proof.* Define $\boldsymbol{v}_{ijk}(\mathbf{X})$ to be the vector whose $L_p$ norm is taken in the computation of $z_{ijk}(\mathbf{X})$. We find

$$z_{ijk}(\mathbf{X} + \mathbf{R}) = \|\boldsymbol{v}_{ijk}(\mathbf{X} + \mathbf{R})\|_p = \|\boldsymbol{v}_{ijk}(\mathbf{X}) + \boldsymbol{v}_{ijk}(\mathbf{R})\|_p$$
$$\le \|\boldsymbol{v}_{ijk}(\mathbf{X})\|_p + \|\boldsymbol{v}_{ijk}(\mathbf{R})\|_p = z_{ijk}(\mathbf{X}) + z_{ijk}(\mathbf{R}).$$

$\square$

**Lemma B.6.** *Average pooling satisfies Assumption 4.3.*

*Proof.* We have

$$z_{ijk}(\mathbf{X} + \mathbf{R}) = \frac{1}{q^2} \sum_{(n,m) \in I} (x_{inm} + r_{inm})$$

$$= \frac{1}{q^2} \sum_{(n,m) \in I} x_{inm} + \frac{1}{q^2} \sum_{(n,m) \in I} r_{inm}$$

$$= z_{ijk}(\mathbf{X}) + z_{ijk}(\mathbf{R}).$$

$\square$

## Footnotes

[1]Note that even though $\mathbf{W} \star (\mathbf{X} + \mathbf{R}) + b$ does not represent a real convolution in this context, the linearity property of Lemma B.1 still applies.