[Reviews · NeurIPS 2017]

Reviewer 1



This paper introduces lower bounds on the minimum adversarial perturbations that can be efficiently computed through layer-wise composition. The idea and the approach is timely, and addresses one of the most pressing problems in Machine Learning. That being said, my main criticism is that the bounds are too loose: the minimum adversarials found through FGSM are several orders of magnitude larger then the estimated lower bounds. That might have two reasons: for one the lower bounds per layer might not be tight enough, or the adversarials found with FGSM are simply to large and not a good approximation for the real minimum adversarials perturbation. To test the second point, I’d encourage the authors to use better adversarial attacks like LBFGS or DeepFool (e.g. using the recently released Python package Foolbox which implements many different adversarial attacks). Also, the histograms in Figure 2&3 are difficult to compare. A histogram with the per-sample ratio between adversarial perturbation and lower bound would be more enlightening (especially once the bounds get tighter).

Reviewer 2



The authors of this manuscript derive the lower bounds on the robustness to adversarial perturbations on feed-foreword neural networks, particularly for CNNs. The theory is validated on MNIST and CIFAR-10 data sets. Overall, the paper is well structured; and the idea is presented well. One major question is how this theory can be extended to GAN network? Moreover, the references at the end of the paper should be rigorously reformatted. I can identify various errors/typos, such as Iclr, mnist, lowercase-uppercase inconsistency, etc.

Reviewer 3



This proposes analysis of the network to bound the size of the adversarial perturbations for deep convolutional networks. The exact shape and reason for adversarial examples is an increasingly more studied domain of neural networks. Robustness/sensitivity to adversarial examples can have various security implications. This paper gives an iterative method for computing the robustness (minimum perturbation necessary to change the output class label) by precomputing the bound on the most commonly occurring layer types (fully connected, convolutional + ReLU, max-pooling) and describing how to back-propagate the estimate. The paper has two main uses: theoretical analyses of networks can be useful for comparing the robustness of different models. Also the bounds could be useful for optimizing the networks for robustness. Although the work is relatively straightforward, it could present a significant milestone toward improved analysis and construction of networks for adversarial robustness.